# UI-Pro: A Hidden Recipe for Building Vision-Language Models for GUI Grounding

## Abstract

Building autonomous UI agents that automate user interactions with interfaces has long been a vision in the field of artificial intelligence. Central to these agents is the capability for UI element grounding, which involves accurately locating UI elements (e.g., buttons and links) based on referring expression, such as user intents and functionality descriptions. Developing these agents with robust grounding capabilities using vision-language models (VLMs) offers a promising path forward. However, a practical framework for creating VLMs with strong element grounding capabilities remains under-explored. To address this gap, we conduct systematic experiments within the design space of VLMs to uncover an effective recipe for building VLMs with strong UI element grounding ability. Firstly, we find that fine-tuning with general visual grounding tasks as a warming-up step mitigates the challenges of fine-tuning with downstream UI element grounding data. Next, we explore different fine-tuning sequences of UI grounding training data from various sources and find that a simple-to-complex fine-tuning curriculum can maximize data utility. Moreover, we find that scaling up the size of either the warming-up data or the UI grounding data in downstream fine-tuning significantly enhances UI element grounding accuracy. Lastly, we explore various image feature compression techniques and find that using a convolution-based compressor to compress UI sub-image features significantly enhances the grounding capabilities on high-resolution UI images. Integrating these insights, we successfully develop UI-Pro, an expert VLM that achieves state-of-the-art UI grounding accuracy with fewer parameters across multiple benchmarks. We hope this work serves as a valuable roadmap for researchers in the UI-VLM domain and inspires future research.

## 1 Introduction

The concept of autonomous UI agents capable of clicking, typing, and scrolling on behalf of humans as personal assistants is an enticing prospect, as illustrated in Fig. 1. Imagine a UI agent navigating the Internet to perform daily tasks such as using search engines and managing emails, as well as more complex activities like comparing prices across e-commerce platforms and collecting the latest news on stock markets.

At the core of autonomous UI agents is *UI element grounding*, which involves recognizing and locating elements associated with referring expressions. These elements serve as the fundamental building blocks that carry UI functionalities. Accurate grounding allows UI agents to interact effectively with UI components such as buttons, text fields, and images, enabling them to perform tasks like clicking, filling out forms, and extracting information according to user instructions. Developing these agents based on vision-language models (VLMs) (Yin et al., 2023) offers a promising pathway toward realizing this vision. A VLM-based UI agent can directly perceive and interact with UIs as humans do, provided the agent possesses vision and comprehension capabilities that align with those of humans. Although a few prior studies, such as SeeClick (Cheng et al., 2024) and CogAgent (Hong et al., 2023) have explored developing UI-related VLMs, a practical recipe for building VLMs with robust UI grounding capabilities from scratch remains under-explored. Specifically, it is unclear which combination of data is most effective for instilling UI grounding capabilities in VLMs and whether non-UI-related multimodal understanding tasks can serve as useful training data. Furthermore, the optimal design of model architectures and training procedures to enhance the models' ability to perform element grounding on high-resolution UI screenshots is still uncertain. Addressing

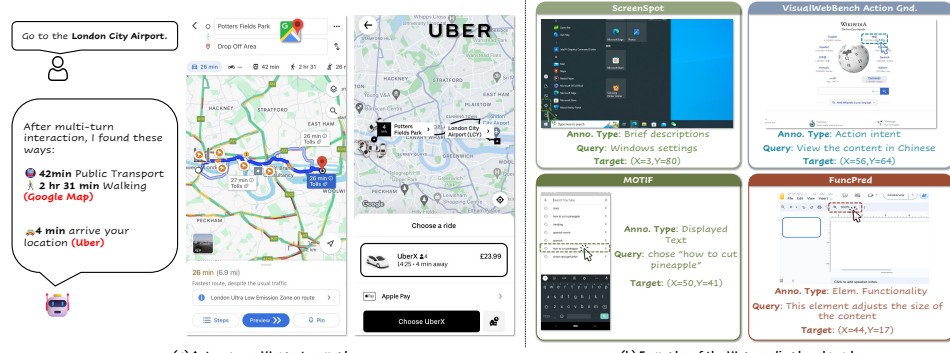

Figure 1: Examples of an autonomous UI agent (left) and existing benchmarks evaluating UI grounding performance (right).

these questions necessitates thorough exploration within the vast design space of VLMs. While several studies (Tong et al., 2024; McKinzie et al., 2024; Laurençon et al., 2024) have examined the significance of various model components and data choices, they predominantly focus on visual question answering (VQA) in natural images, overlooking the complex challenges presented by UI grounding scenarios.

This paper aims to bridge the gap in existing research by providing a practical framework for building VLMs capable of UI grounding. Drawing on pioneering research (Tong et al., 2024; McKinzie et al., 2024; Baechler et al., 2024), we pinpoint three key areas where different studies make distinct design choices: (a) model architecture, particularly vision-language connection modules that enhance the accuracy of locating small elements within high-resolution UI screenshots, presenting new challenges rarely addressed in visual grounding for natural images; (b) training data curation; and (c) fine-tuning procedures. We systematically compare various design choices in a controlled setting to derive empirical insights for each area.

Our findings reveal that: (a) warming up VLMs with general visual grounding task data is essential before fine-tuning the models on downstream UI grounding tasks; (b) organizing UI grounding training data in a simple-to-complex curriculum significantly maximizes data utility through multi-stage fine-tuning; (c) increasing the sizes of both the warming-up data and UI grounding data results in substantial performance gains, well beyond saturation; and (d) a lightweight convolution-based connector is effective for compressing visual features of high-resolution UI images, enabling processing at the original ratio.

Our work distinguishes itself from previous studies (Yao et al., 2022; Cheng et al., 2024; Hong et al., 2023; You et al., 2024b) on UI-oriented VLMs by exploring previously unexamined areas, including warming-up data selection, multi-stage training methodologies, data scaling effects, and UI-oriented connector design. Our findings provide a hidden recipe for building powerful UI VLMs from scratch, circumventing reliance on fine-tuning open-source models.

Based on these insights, we have trained UI-Pro, a VLM with 2.8 billion parameters. UI-Pro demonstrates exceptional performance across multiple UI grounding benchmarks, including grounding by action intents, element appearance, and complex functionality descriptions. Notably, UI-Pro matches the performance of previous UI-oriented models that are nine times its size. We hope our work will benefit the research community and accelerate advancements in UI autonomous agents.

## 2 PRELIMINARIES

### 2.1 UI ELEMENT GROUNDING TASK

UI element grounding is to locate visual elements within UIs given element annotations. These annotations can be brief, including details such as element appearance, location, and displayed text, or complex, encompassing contextual functionality and action intents, as shown in Fig. 1.

Several element grounding benchmarks have been established for research purposes (Fig 1). **ScreenSpot** (Cheng et al., 2024) is a benchmark for mobile, desktop, and web scenarios, requiring models to locate elements based on brief descriptions. **RefExp** (Bai et al., 2021) focuses on locating elements on mobile devices using crowd-sourced referring expressions. **VisualWebBench** (Liu et al., 2024c) evaluates VLMs in content-rich web environments. In contrast to these benchmarks, **AutoGUI Test (FuncPred)** features complex tasks that require models to locate elements specified by context-specific functionality descriptions. For all these benchmarks, we report the grounding accuracy (%): $\mathrm{Acc} = \sum_{i=1}^{N} \mathbf{1}\left(\mathrm{pred}_i \text{ inside GT bbox}_i\right)/N \times 100$

where $\mathbf{1}$ is an indicator function and $N$ is the number of test samples. This formula denotes the percentage of samples for which the predicted points lie within the bounding boxes of the elements.

Unlike visual grounding aimed at natural scenes (Yu et al., 2016), UI element grounding introduces distinct challenges: 1) High resolution: UIs are typically rendered as high-resolution images, necessitating models that can process large visual inputs effectively. 2) Fine-grained comprehension: UIs often display numerous small elements, which occupy significantly less area than objects in datasets like RefCOCO (Yu et al., 2016) and Visual Genome (VG) (Krishna et al., 2016), requiring enhanced visual understanding to distinguish between highly similar elements. 3) Data insufficiency. Due to the substantial cost of human annotation, the scale of existing open-source datasets for UI understanding is significantly lower than natural image datasets such as COCO (Lin et al., 2014) and LAION-5B (Schuhmann et al., 2022).

## 2.2 BASICS OF VLMS

We adopt the popular architecture used by recent VLMs, such as LLaVA (Liu et al., 2023) and Qwen-VL (Bai et al., 2023). These architectures typically combine a pre-trained visual backbone $f_\phi$ (e.g., ViT (Dosovitskiy et al., 2021)) and a large language model (e.g., Llama Touvron et al. (2023)) to build a model capable of processing both textual and visual inputs. Formally. the visual backbone maps an input image $\mathbf{l} \in \mathbb{R}^{H \times W \times 3}$ to an $L$-length sequence of patch features $V_{img} \in \mathbb{R}^{L \times h}$ that are then projected into the embedding space of the LLM. Subsequently, the projected visual features $E_{img} \in \mathbb{R}^{L \times D}$ are concatenated with the $S$-length textual embeddings $E_{txt} \in \mathbb{R}^{S \times D}$ before being fed to the LLM for response generation. The generation process can be formulated as $o = \mathrm{LLM}_\theta([Proj_\omega(f_\phi(\mathbf{l})), Embed(\boldsymbol{t})]$; where $\boldsymbol{t}$, $Proj$, and $Embed$ denotes the text prompt, the vision-language projector, and the embedding module in the LLM, respectively. Given a training sample ($\mathbf{l}$, $\boldsymbol{t}$, $o$), the VLM is optimized by minimizing the loss $L(\phi, \omega, \theta) = -\log p(o|\mathbf{l}, \boldsymbol{t})$ via gradient descent.

In this paper, we aim to explore the intricate design space of VLMs to develop a comprehensive recipe for building VLMs capable of UI element grounding.

## 2.3 DOWNSTREAM UI ELEMENT GROUNDING TRAINING DATASETS

To fulfill our aim, two training datasets are utilized as the sources for downstream fine-tuning:

**SeeClick** (Cheng et al., 2024) provides a dataset that integrates existing tasks in UI element grounding, captioning, and summarization. It comprises two parts: (a) a web portion containing element text grounding and OCR tasks derived from 300k web pages in the latest Common Crawl repository[1]; and (b) a mobile device portion that includes element grounding and captioning tasks generated by applying instruction-following templates to the Widget Captioning and RICO annotations. As shown in Fig. 1, this dataset mainly comprises brief element annotations.

**AutoGUI** (AutoGUITeam, 2024) is introduced to complement SeeClick. AutoGUI contains 625k UI functionality grounding and captioning tasks that require VLMs to grasp the functional semantics of various UI elements. This dataset is collected on multi-resolution and multi-device screenshots across diverse data domains, providing detailed element functionality annotations related to UI contexts (see Fig. 1). Given that the functionality annotations in this AutoGUI dataset are more detailed and longer than those in SeeClick and that associating these annotations with unique elements among hundreds of counterparts is challenging, this dataset is expected to enhance VLMs' UI element grounding capabilities, albeit with increased complexity.

---

[1] https://index.commoncrawl.org/

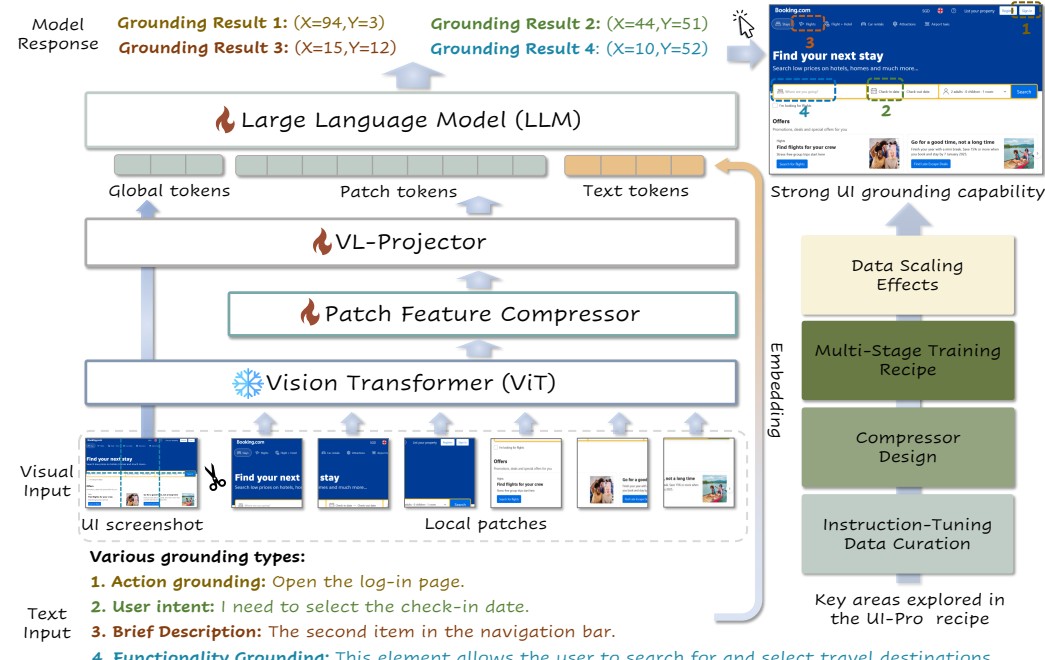

Figure 2: The UI-Pro pipeline and training recipe. The recipe includes (a) Data Curation, (b) Compressor Design, (c) Training Recipe, and (d) Data Scaling for enhancing UI grounding capabilities, as explored in this work. Extensive experiments (conducted in this paper) demonstrate that constructing datasets for warm-up phases, curriculum-based training, scaling datasets, and employing appropriate compression modules play crucial roles in enhancing UI-element grounding performance.

## 3 EXPLORING THE DESIGN SPACE OF VLMS FOR IMPROVED UI GROUNDING

This section investigates design choices related to instruction-tuning data, training procedures, and patch feature compression techniques for UI VLMs. A LLaVA model (Liu et al., 2023) with a pre-trained vision-language projector is utilized as the base model. To process high-resolution UI screenshots, a parameter-free image division strategy (Ye et al., 2023a; Zhang et al., 2024b) is employed to crop the shape-variable screenshots into fixed-size image patches. Please see Fig. 2 for the full pipeline. More implementation details are listed in the Appendix.

### 3.1 WHICH DATA TYPE CAN BE USED TO WARM UP VLMS?

Our preliminary studies found that directly fine-tuning the base VLM using UI element grounding data resulted in poor grounding accuracy. We hypothesize that the base model needs warming up to adapt to UI element grounding tasks, which require models to output precise numerical coordinates based on cluttered UI screenshots.

To explore suitable warming-up data sources, multiple instruction-tuning tasks shown in Fig. 3 are collected for comparison:

**Visual Grounding on Natural Images** requires VLMs to output the bounding boxes of target objects given the object descriptions. We convert the bounding box annotations in RefCOCO (Yu et al., 2016), RefCOCO+ (Yu et al., 2016), RefCOCOg Nagaraja et al. (2016), and VG (Krishna et al., 2016) into visual grounding and referring tasks by applying instruction-tuning templates, resulting in 5.7M samples in total.

**Question Answering on Natural Images** involves generating natural language responses by following text instructions, without coordinate outputs. We utilize the VQA subset, including LLaVA-Pretrain (Liu et al., 2023), COCO Lin et al. (2014), and SAM (Kirillov et al., 2023), of the ShareGPT4V-SFT dataset (Chen et al., 2023a), resulting in 530k samples in total.

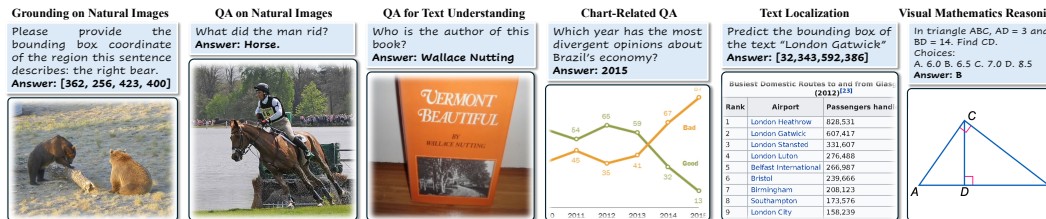

Figure 3: Examples of the task types tested and compared in the warming-up stage.

Table 1: **Evaluating the base model warmed up with various tasks.** The base model is first fine-tuned using a warming-up task and then trained on the downstream AutoGUI task. We can see that visual grounding on natural images significantly enhances accuracy grounding accuracy. Text localization also yields high accuracy. Text-QA, Chart-related QA, and Visual mathematical reasoning tasks provide minimal benefit. VQA on natural images is also not pretty useful. Although ShareGPT4V-SFT includes a variety of tasks, it performs worse than using either pure natural image grounding or text localization task data. Notably, directly fine-tuning with AutoGUI data without warming up results in inferior performance.

| Warming-up Task | FuncPred | ScreenSpot | MOTIF | RefExp | VWB EG | VWB AG |
|---|---|---|---|---|---|---|
| Gnd. on Natural Images | 42.6 | **19.4** | **28.3** | **11.5** | **8.5** | **8.7** |
| QA on Natural Images | 40.2 | 12.8 | 19.8 | 9.6 | 5.3 | 1.9 |
| Text Localization | **46.3** | 12.5 | 24.3 | 8.1 | 4.8 | 2.9 |
| Text-QA | 39.0 | 10.4 | 21.1 | 9.2 | 3.6 | 1.0 |
| Chart-Related QA | 36.2 | 12.3 | 16.4 | 10.8 | 3.6 | 1.9 |
| Visual Math. Reasoning | 37.0 | 7.9 | 15.2 | 5.7 | 0.7 | 0.0 |
| ShareGPT4V SFT | 35.7 | 7.5 | 9.2 | 7.6 | 5.1 | 3.9 |
| None | 35.2 | 4.2 | 9.6 | 1.2 | 1.7 | 1.0 |

**Question Answering for Text Understanding (Text-QA)** focuses on recognizing and interpreting textual contents in images to answer questions. We extract the combination of the TextVQA (Singh et al., 2019), ShareTextQA, and OCR-VQA (Mishra et al., 2019) portions from the ShareGPT4V-SFT dataset, resulting in 102k samples in total.

**Text Localization** combines text grounding and recognition tasks, requiring the prediction of bounding boxes for situated texts and their recognition. We use the 1M text localization subset of the DocStruct4M data proposed by mPLUG-DocOwl-1.5 (Hu et al., 2024).

**Visual Mathematics Reasoning** requires VLMs to understand complex mathematical diagrams and formulas for multi-modal reasoning. We combine Inter-GPS (Geometry Problem Solving) (Lu et al., 2021), GeoQA Chen et al. (2021), and MATH-Vision (Wang et al., 2024a), obtaining 82.5k samples.

**Chart-Related Question Answering** tasks VLMs with answering questions about data visualizations, e.g. scientific diagrams and statistical tables from textbooks and academic papers, challenging visual and logical reasoning over charts. We collect data from ArXivQA Li et al. (2024a), ChartQA (Masry et al., 2022), ScienceQA (Lu et al., 2022), TabMWP (Lu et al., 2023), TextbookQA (Kembhavi et al., 2017), AI2D (Kembhavi et al., 2016), and DVQA (Kafle et al., 2018), curating 394k samples in total.

**ShareGPT4V SFT** is also used to explore whether combining various types is beneficial. This dataset contains 665k samples, including VQA, visual grounding, and Text-QA tasks.

We restrict the number of samples to 355k by randomly sampling from tasks with more than 355k and resampling those with fewer. This experiment follows a two-stage fine-tuning approach: the base model is first fine-tuned with warming-up tasks, followed by fine-tuning with 125k samples from the AutoGUI dataset. Tab. 1 shows that visual grounding on natural images as the warming-up task significantly enhances accuracy accuracy on the UI grounding benchmarks. Text localization also achieves high accuracy on FuncPred but performs poorly on ScreenSpot. Although text QA, chart-related QA, and visual mathematical reasoning tasks are aimed at enhancing the fine-grained understanding capabilities of VLMs, their overall gains are limited. Interestingly, ShareGPT4V-SFT, which includes diverse tasks, does not provide benefits in the downstream fine-tuning stage for the

Table 2: **Experiments on the fine-tuning curriculum of the warming-up, SeeClick, and AutoGUI datasets.** The fine-tuning procedure is divided into three stages, each of which fine-tuning the base model with a different dataset. Notably, fine-tuning with SeeClick containing simple UI grounding tasks and then with AutoGUI containing complex functionality grounding tasks contributes to high accuracy over the two benchmarks (row 6). Reversing or mixing the two UI datasets leads to deteriorated performance on FuncPred (rows 7 and 8). These results indicate that organizing the two UI grounding datasets with a simple-to-complex curriculum helps to maximize data utility.

| No. | SFT-1 | SFT-2 | SFT-3 | FuncPred | ScreenSpot |
|-----|-------|--------|--------|----------|------------|
| r1 | - | SeeClick | - | 17.3 | 39.9 |
| r2 | - | - | AutoGUI | 46.3 | 14.3 |
| r3 | - | SeeClick | AutoGUI | 56.7 | 41.1 |
| r4 | Gnd. | SeeClick | - | 20.8 | 44.0 |
| r5 | Gnd. | - | AutoGUI | 52.0 | 24.9 |
| r6 | Gnd. | SeeClick | AutoGUI | **57.7** | **44.7** |
| r7 | Gnd. | AutoGUI | SeeClick | 22.3 | 43.9 |
| r8 | Gnd. | - | SeeClick+AutoGUI | 52.0 | 44.4 |

UI dataset, suggesting that this diverse dataset is not an ideal warming-up source for enhancing UI element grounding capabilities, despite its common use in existing VLM studies (McKinzie et al., 2024; Tong et al., 2024). In summary, these results demonstrate that grounding tasks from either the natural image or text-rich scenarios serve as desirable warming-up data sources.

> ***Finding* 1.** Directly fine-tuning VLMs with UI element grounding data can lead to training difficulty. Utilizing visual grounding tasks from either natural or text-rich scenarios as a warming-up step significantly enhances downstream fine-tuning with UI element grounding tasks.

## 3.2 What Fine-tuning Curriculum Can Maximize Dataset Utility?

Given a warming-up dataset, a UI grounding dataset with simple annotations (SeeClick), and one with complex functionality semantics (AutoGUI), a question arises of how we can optimize the fine-tuning order to fully leverage the advantages of these datasets. In this experiment, we aim to find a suitable fine-tuning procedure for utilizing datasets from various sources to enhance VLMs' UI element grounding capability. We explore different fine-tuning orders of the three datasets and compare the performances. Specifically, the fine-tuning process is divided into three stages: the first stage uses visual grounding on natural images (355k samples) to warm up the base model according to the finding in Sec. 3.1; the second uses 355k samples from the simple tasks in SeeClick; the third uses the 625k complex tasks in AutoGUI. Each stage is run for one epoch.

The results in Tab. 2 demonstrate that initial fine-tuning with the warming-up task consistently enhances downstream fine-tuning with UI grounding data (r4 vs. r1, r5 vs. r2, and r6 vs. r3), especially when the downstream task is the hard functionality grounding task of AutoGUI. This observation aligns with findings in Sec. 3.1. Fine-tuning with first SeeClick (simple) and then AutoGUI (complex) generally yields better performance than variants that fine-tune exclusively with either SeeClick or AutoGUI (r3 vs. r1 and r2). This trend persists even when models are warmed up, obtaining accuracy gains of 5.7 and 18.8 on the FuncPred and ScreenSpot, respectively (r6 vs. r5). An exception appears when comparing r6 and r4, where performance on ScreenSpot remains unchanged despite a significant increase on FuncPred, likely due to a small domain gap between the functionality grounding task of AutoGUI and the grounding-by-brief-descriptions task of the ScreenSpot benchmark.

Interestingly, reversing the order of SeeClick and AutoGUI (r7 vs. r6) leads to a significant performance drop of 35.4 on AutoGUI. Additionally, mixing these tasks for fine-tuning leads to a decrease in FuncPred (r8 vs. r6). These findings indicate that the base model requires warming up with the simpler UI grounding task before fine-tuning with more complex tasks. In summary, these results indicate that organizing the three datasets in a simple-to-complex curriculum maximizes data utility.

> ***Finding 2.*** In the downstream fine-tuning stage after warming up, structuring UI grounding task datasets in a simple-to-complex curriculum significantly enhances data utility.

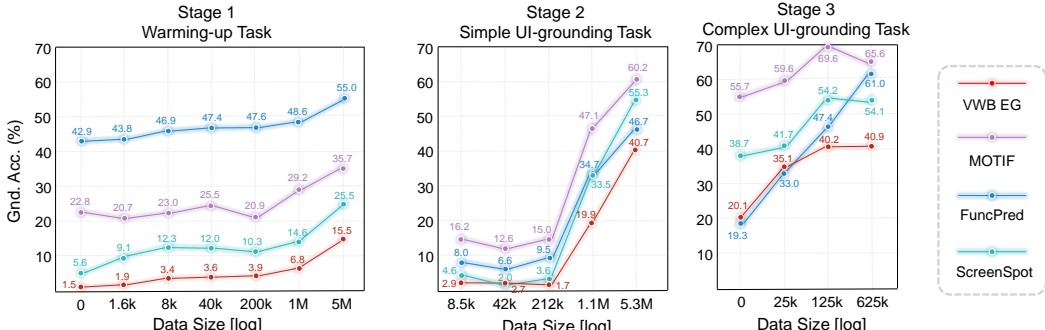

Figure 4: **Scaling effects for the warming up, simple UI grounding (SeeClick), and complex UI grounding (AutoGUI) data.** This figure highlights that increasing the warming-up and UI grounding data remarkably improves performance. Scaling the AutoGUI data shows modest gains, with peak performance observed at 125k samples for MOTIF and ScreenSpot, suggesting potential overfitting.

### 3.3    WHAT IS THE SIGNIFICANCE OF FINE-TUNING DATA SIZE?

Exploring the effects of data size scaling in training VLMs is pivotal for optimizing performance as increased data has been observed to lead to enhanced model generalization and performance (Kaplan et al., 2020; Brown et al., 2020; Zhao et al., 2023; Liu et al., 2024a; Karamcheti et al., 2024).

We systematically assess how varying fine-tuning data sizes impact the performance of VLMs in the UI element grounding tasks. Following the insight in Sec. 3.2, we adopt a three-stage fine-tuning procedure, employing the warming up, SeeClick, and AutoGUI data in stages 1, 2, and 3, respectively.

**Scaling warming-up data in stage 1.**    Inspired by the power-law scaling observed in (Kaplan et al., 2020), we scale the visual grounding on natural image data, as discussed in Sec. 3.1, across seven levels: 0, 1.6k, 8k, 40k, 200k, 1M, and 5M samples. 212k samples of SeeClick data are used in stage 2 and 625k AutoGUI data are used in stage 3.

**Scaling simple UI-grounding data in stage 2.**    The SeeClick training data is scaled across five levels: 8.5k, 42k, 212k, 1.06Mk, and 5.3M samples. 5M samples of the warming-up data are used in stage 1 and 625k AutoGUI data are used in stage 3.

**Scaling complex UI-grounding data in stage 3.**    The AutoGUI training data is extracted and scaled across four levels: 0, 25k, 125k, and 625k samples. This experiment utilizes 5M samples of the warming-up data and 212k samples of SeeClick data.

The results in Fig. 4 show that scaling up the warming-up data in stage 1 contributes to significant improvements in benchmark performance, even though the domain of this data (natural images) differs from the UI-specific data used in subsequent fine-tuning stages. This suggests that the model acquires a preliminary capability of fine-grained spatial localization and numerical coordinate generation, which are essential for tackling the more challenging UI grounding tasks downstream. Scaling up SeeClick data in stage 2 also brings significant performance gains across all the benchmarks, with the scale of 212k serving as a critical reflection point. Scaling the AutoGUI data in stage 3 results in modest improvements, peaking at 125k on MOTIF and ScreenSpot. The FuncPred accuracy continues to rise, likely due to its alignment with the AutoGUI task domain, while performance on the other benchmarks plateaus or slightly declines, possibly indicating overfitting to the AutoGUI task format. In summary, while scaling the warming-up and UI grounding data enhances performance, attention should be paid to the risk of overfitting.

***Finding* 3.** Scaling warming-up data and UI grounding data significantly enhances element grounding accuracy. It is also important to remain cautious of potential overfitting when finetuning with complex UI grounding data.

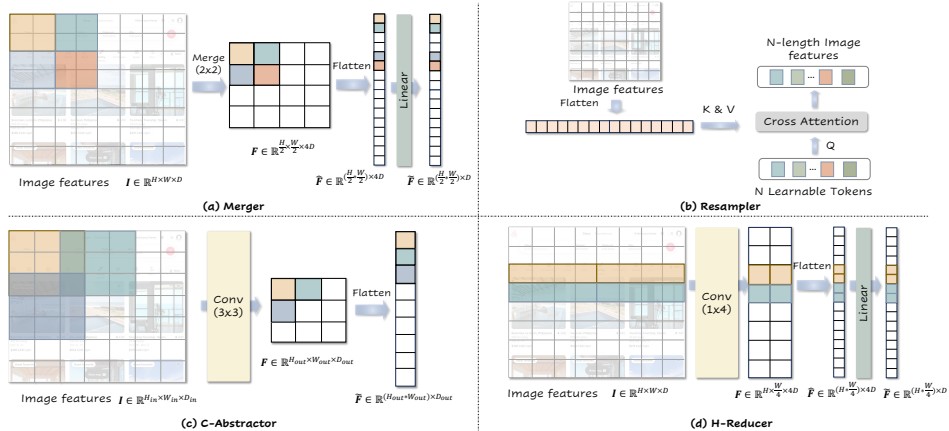

Figure 5: **Comparing the patch feature compressors.** (a) Merger concatenates the nearby 4 tokens into a new token along the channel dimension and then reduces the channel dimension to 1/4 with an MLP. (b) Resampler uses a fixed set of learnable latent queries that interact with each patch feature through cross-attention, outputting a fixed-length visual feature. (c) C-Abstractor employs a traditional convolution network to compress patch features. (d) H-Reducer uses a convolution network whose kernel size and stride size are set as $1 \times 4$ to fuse horizontal 4 visual tokens.

### 3.4 WHICH PATCH FEATURE COMPRESSION APPROACH IS BENEFICIAL?

UIs captured at super-high resolutions result in thousands of visual tokens when processed with the division technique. For example, a 720p screenshot will be converted into 4608 tokens and a 4kHD will exceed 48k. These extensive features may possess redundant details that interfere with VLM inference and cause an unbearably high computational budget.

To build a UI VLM capable of efficiently processing high-resolution screenshots, we explore various designs of the patch feature compressor, as shown in Fig. 5: (a) **Merging compressor (Merger)**(Ye et al., 2023a): This compressor concatenates $N = 4$ adjacent tokens in square regions along the channel dimension andcompresses the concatenated features using a single-layer MLP. (b) **Resampler** (Alayrac et al., 2022): This compressor reduces visual features to a fixed number of tokens by utilizing a set of learnable queries to cross-attend to the visual features. (c) **Convolution-based compressor**: This compressor reshapes the visual features to align them with image dimensionality, processes them through a convolutional network, and flattens them back into visual tokens. Apart from convolution with square-shaped kernels, i.e., **C-Abstractor** (Cha et al., 2024), we also test the **H-reducer** (Dong et al., 2024), a type with stripe-shaped kernels ($1 \times 4$), which is tailored for horizontal text layouts in document understanding scenarios. (d) **MLP**: Directly using an MLP to process the patch features without compression.

This experiment uses 1M warming-up samples, 1.1M SeeClick data, and 125k AutoGUI data to fine-tune the base model for one epoch. To ensure a fair comparison, the numbers of parameters of these compressors are roughly equalized by adjusting their hyper-parameters.

The results in Tab. 3 show that the two convolution-based compressors achieve superior grounding accuracy, with C-Abstractor leading on five benchmarks. Although H-Reducer is tailored for document inputs, it is inferior to C-Abstractor which uses square-shaped kernels, probably because the UI screenshots display a higher proportion of flexibly arranged visual contents (i.e., icons and images), compared to text-rich documents. Resampler performs poorly as its cross-attention mechanism possibly leads to a loss of spatial information, which is crucial for grounding tasks that require precise annotation-region associations (Cha et al., 2024). In contrast, Merger simply fuses nearby

Table 3: **Comparing the performances of introducing different patch feature compressors.** The convolution-based compressor obtains higher accuracy than the other types, with C-Abstractor exhibiting leading performances on five benchmarks. Without a compressor, the model can only employ the low-resolution raw image, resulting in significantly inferior performance.

| Compressor Type | FuncPred | ScreenSpot | MOTIF | RefExp | VWB EG | VWB AG |
|---|---|---|---|---|---|---|
| None | 20.1 | 3.8 | 18.0 | 4.2 | 1.5 | 1.0 |
| MLP (Liu et al., 2024b) | 26.2 | 13.6 | 36.1 | 20.0 | 7.0 | 20.4 |
| Merger (Ye et al., 2023a) | 35.9 | 38.5 | 51.3 | 26.0 | 16.0 | 26.2 |
| Resampler (Alayrac et al., 2022) | 35.1 | 29.2 | 50.3 | 21.6 | 13.6 | 13.6 |
| C-Abstractor (Cha et al., 2024) | 34.3 | **42.1** | **58.0** | **32.0** | **16.5** | **30.1** |
| H-Reducer (Dong et al., 2024) | **37.0** | 39.7 | 55.3 | 28.3 | 10.2 | 27.2 |

Table 4: **Comparing UI-Pro with leading VLMs on the UI element grounding benchmarks.** The results demonstrate that UI-Pro achieves impressive grounding accuracy with much fewer parameters. AnyRes means that the method uses an image division strategy to handle images with variable resolutions.

| Model | Size | Input Res. | FuncPred | ScreenSpot | MOTIF | RefExp | VWB EG | VWB AG |
|---|---|---|---|---|---|---|---|---|
| LLaVA-1.5 (Liu et al., 2024a) | 7B | 336 | 3.2 | 5.0 | 7.2 | 4.2 | 12.1 | 13.6 |
| LLaVA-1.5 (Liu et al., 2024a) | 13B | 336 | 5.8 | 11.2 | 12.3 | 20.3 | 16.7 | 9.7 |
| LLaVA-1.6 (Liu et al., 2024b) | 34B | AnyRes | 4.4 | 10.3 | 7.0 | 29.1 | 19.9 | 17.0 |
| SliME (Zhang et al., 2024b) | 8B | AnyRes | 3.2 | 13.0 | 7.0 | 8.3 | 6.1 | 4.9 |
| MiniCPM-V-2.6 (Yao et al., 2024) | 8B | AnyRes | 16.5 | 33.0 | 12.9 | 29.3 | 9.4 | 21.7 |
| Qwen-VL (Bai et al., 2023) | 10B | 448 | 3.0 | 5.2 | 7.8 | 8.0 | 1.7 | 3.9 |
| Qwen2-VL (Wang et al., 2024b) | 7B | AnyRes | 7.8 | 26.1 | 16.7 | 32.4 | 3.9 | 3.9 |
| CogAgent (Hong et al., 2023) | 18B | 1120 | 29.3 | 47.4 | 46.7 | 35.0 | **55.7** | **59.2** |
| SeeClick (Cheng et al., 2024) | 10B | 448 | 19.8 | 53.4 | 11.1 | **58.1** | 39.2 | 27.2 |
| UI-Pro-Gemma-2B (ours) | 2.8B | AnyRes | **46.3** | **56.3** | **64.3** | 44.6 | 43.8 | 33.0 |

four tokens, surpassing Resampler by a large margin. The MLP-based variant and the one without compression both show weak UI grounding ability, suggesting the necessity of a compression module. In summary, these results suggest that convolution-based compressors are suitable for enhancing VLMs' UI grounding capabilities.

> ***Finding* 4.** Employing compression modules to reduce patch features is crucial for VLMs that utilize an image division strategy. The use of convolutional networks to compress local patch features significantly enhances the UI grounding capabilities of VLMs.

## 4 STATE OF THE ART PERFORMANCE OF UI-PRO

Finally, we leverage the findings from the previous experiments to build UI-Pro. We train UI-Pro using Gemma-1.1-2B (Team et al., 2024) and LLaMA-3.2-Instruct-3B AI@Meta (2024) as the base LLM and OpenAI CLIP ViT-L/14@336 (Dosovitskiy et al., 2021) as the visual encoder. The training process begins with 5M samples of visual grounding on natural images for warming up, followed by fine-tuning with 5.3M SeeClick and 125k AutoGUI samples, adhering to a simple-to-complex curriculum. We utilize C-Abstractor as the patch feature compressor.

Tab. 4 show that compared with existing VLMs, UI-Pro exhibits impressively high accuracy on the UI element grounding benchmarks. Notably, it achieves this with only one-fifth the model size of CogAgent, a leading UI-oriented VLM, surpassing it across five benchmarks and establishing a new state-of-the-art. In contrast, VLMs primarily designed for universal multimodal comprehension, such as Qwen-VL, MiniCPM-V-2.6, and LLaVA-1.6, struggle with UI element grounding tasks, highlighting a potential disconnect between their design strategies and the complexities of UI grounding scenarios. Overall, with our four key findings, we can build VLMs that possess strong UI element grounding capabilities.

## 5 RELATED WORKS

### 5.1 RECENT ADVANCEMENT OF VLMS

There has been a significant rise in reasearch focused on improving LLMs by integrating both visual and textual data (Alayrac et al., 2022; Chen et al., 2023b; Li et al., 2023; Lin et al., 2023a; Liu et al., 2023; Lin et al., 2023b; Chen et al., 2023c; Lu et al., 2024; Bai et al., 2023; Wang et al., 2024b; Zhu et al., 2024; Wang et al., 2024c; Li et al., 2024b; Zhang et al., 2024a; You et al., 2024a; Laurençon et al., 2024; Peng et al., 2024; Driess et al., 2023), which has led to the development of VLMs. Flamingo (Alayrac et al., 2022), utilizes combined visual and language prompts and has demonstrated exceptional few-shot visual question-answering abilities. With the advancements brought by GPT-4 (Team, 2024), both academic and industrial efforts have been made to make its multimodal reasoning capabilities more accessible. LLaVA (Liu et al., 2023) and LLaMA-Adapter (Zhang et al., 2024a) have worked to align vision encoders (Dosovitskiy et al., 2021) with LLMs to support visual instruction following. Models like VisionLLM (Wang et al., 2024c), Ferret (You et al., 2024a), and Qwen-VL (Bai et al., 2023) have further developed strong visual grounding abilities. LLaVA-Next Liu et al. (2024b), Monkey Li et al. (2024b), LLaVA-UHD Guo et al. (2024), Qwen2-VL Wang et al. (2024b) enhanced the perception resolution of VLMs. Moreover, research is expanding to VLM applications in contexts with rich textual imagery (Tang et al., 2022; Ye et al., 2023b;a; Liu et al., 2024d) and embodied interactions (Driess et al., 2023; Mu et al., 2023), unlocking new possibilities in multimodal reasoning. Additionally, some works Laurençon et al. (2024); McKinzie et al. (2024) give a comprehensive study on building VLMs, highlighting the impact of various design components and data choices on model performance. Despite these advancements, there has been no systematic approach proposed for data collection, model design, or training frameworks specifically targeting VLMs in UI environments, highlighting a critical gap in the research.

### 5.2 EXISTING UI DATASETS AND BENCHMARKS

In contrast to well-established natural image datasets (Russakovsky et al., 2014; Schuhmann et al., 2022), datasets focused on UI understanding have received less attention in the field of computer vision. Some efforts have been made to develop datasets for mobile UI modeling (Wang et al., 2021; Li et al., 2020a;b; Bai et al., 2021; Burns et al., 2022), with many of these efforts centered on the RICO dataset (Deka et al., 2017), which contains 72K Android app screenshots. Notable examples include Widget Captioning (Li et al., 2020a), which examines the captions and linguistic features of UI elements, and RICOSCA (Li et al., 2020b), which maps single-step instructions to corresponding UI elements. More recently, MoTIF (Burns et al., 2022) has gained attention alongside the growing interest in web-based scenarios. WebShop (Yao et al., 2022), for instance, was an early effort to introduce a simplified simulator for web navigation tasks. Subsequent projects like Mind2Web (Deng et al., 2024) and WebArena (Zhou et al., 2023) have focused on creating realistic and reproducible web environments to enhance web agent performance. VisualWebBench (Liu et al., 2024c) has also contributed by establishing a robust evaluation framework for VLMs, specifically targeting UI grounding. To address the issue of limited data, recent studies like SeeClick (Cheng et al., 2024) and CogAgent (Hong et al., 2023) have leveraged Common Crawl data to construct large-scale datasets, though these datasets often include noisy HTML code snippets. AITW (Rawles et al., 2023) has been introduced to focus on interpreting high-level instructions in Android environments. Existing UI-VLMs have primarily focused on fine-tuning open-source models using these datasets, but there is still a lack of detailed solutions for effectively enhancing their UI grounding capabilities.

## 6 CONCLUSION

This paper introduces a practical framework for building VLMs with strong UI element grounding capability. We identified key strategies, including warming up with visual grounding tasks, employing a simple-to-complex fine-tuning curriculum, and scaling data sizes, all of which significantly optimize grounding accuracy. Our findings on image feature compression further improve grounding accuracy for high-resolution UI images. The integration of these findings resulted in UI-Pro, a state-of-the-art VLM that achieves impressive grounding accuracy with fewer parameters. Hope this research provides a roadmap for future studies in intelligent UI agent development.

REPRODUCIBILITY STATEMENT

UI-Pro is fully reproducible. The fine-tuning code is based on the open-source LLaVA repo and all the used training data in the experiments are also open-sourced. As the four findings proposed in the paper are easy to put into practice, readers can reproduce our results by modifying LLaVA repo according to the model designs Fig. 5, fine-tuning curriculum in Sec 3.2, and hyperparameter settings in the Appendix.

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

Table A: The training hyper-parameters used for fine-tuning UI-Pro in the experiments.

| Hyper-Parameter | Value |
|---|---|
| Epoch | 1 |
| Global batch size | 128 |
| #GPUs | 8 |
| Learning rate for all stages | 3e-5 |
| weight decay | 0.0 |
| ADAM Beta2 | 0.95 |
| Warm-up ratio | 0.03 |
| LR scheduler | Cosine |
| Model max length | 2048 |
| Frozen module | ViT |
| DeepSpeed | ZeRO-2 |
| Data type | BFloat16 |

# A APPENDIX

## A IMPLEMENTATION DETAILS

### A.1 DATA CURATION DETAILS

We found that certain datasets, such as RefCOCO and SeeClick include samples containing multi-turn dialogues while others do not, leading to potential imbalance issues caused by this multi-turn trait and resulting in unfair comparison. Additionally, excessively long dialogs exceed the context window of 2048 of UI-pro, causing training bugs. To resolve these issues, we reorganize the multi-turn dialogs to ensure each dialog contains no more than 650 text tokens to balance all samples.

### A.2 FINE-TUNING DETAILS

The hyper-parameters of training UI-Pro is shown in Tab. A. All experiments are conducted with 8 L20 GPUs each with 48GB memory. The three-stage fine-tuning (stage 1: warming-up; stage 2: simple UI grounding task fine-tuning; stage 3: complex UI grounding task fine-tuning) spend approximately 22 hours in total.

Although the parameter numbers of the compressors can not be flexibly adjusted due to discrete parameter space and hardware efficiency issues, we try our best to match their sizes: (a) Merger: 10,488,832, (b) Resampler: 8,998,912, (c) C-Abstractor: 9,100,032, (d) H-Reducer: 10,489,344, (e) MLP: 8,392,704.

## B LIMITATIONS

Despite the impressive UI element grounding capability, UI-Pro still encounters several limitations:

**1. Model Diversity.** This paper is targeted at LLaVA-like architectures that typically comprise a vision encoder, a vision-language connector, and an LLM. In practice, there exist various VLM architectures, such as Flamigo (Alayrac et al., 2022) and Emu (Sun et al., 2024) with multi-modal inputs and outputs. Future work can extend our findings to these architectures to generalize the insights.

**2. UI Data Diversity.** As UI-related datasets are much more scarce than natural image datasets, this work mainly conducts experiments with SeeClick and AutoGUi training datasets, which are the largest ones to date. This data insufficiency issue may be the cause of slight over-fitting observed in the data size scaling experiments, as shown in Fig 4. We hope future work can collect more diverse data to consolidate our findings.

**3. Resource Intensiveness.** Fine-tuning VLMs like UI-Pro can be extremely resource-intensive, requiring substantial computational power and time, which may limit accessibility for some researchers or developers. Due to such a resource restriction, this work uses small LLMs, i.e. Gemma-2B. We will extend our work to a larger scale by integrating larger LLMs like Llama-3-8B.

