# OpenReview forum: "UI-Pro: A Hidden Recipe for Building Vision-Language Models for GUI Grounding"
_ICLR.cc/2025/Conference — ICLR 2025 Conference Withdrawn Submission_

### Official Review · Reviewer_qBF2 · 2024-10-16

**Soundness:** 2
**Presentation:** 2
**Contribution:** 1
**Rating:** 3
**Confidence:** 4

**Summary:**

This paper introduces four independent strategies to improve the performance of vision-language models for GUI Grounding: (1) warm-up with general visual grounding task data, (2) organizing UI grounding training data in a simple-to-complex curriculum through multi-stage fine-tuning, (3) increasing the sizes of both warm-up and UI grounding data, and (4) using a lightweight convolution-based connector to compress visual features of high-resolution UI images, reducing computational cost to enable processing at the original ratio.

**Strengths:**

1. All proposed findings are well-ablated
2. The proposed UI-Pro model achieves good performance using significantly fewer parameters

**Weaknesses:**

This paper reads like an experimental report, merely listing a series of methods proposed by others and reporting their best results. Hence, the paper suffers from considerable lack of novelty, as curriculum learning is a well-established field, the benefits of scaling data to improve performance are well-known, and the convolution-based connectors are also directly adopted from existing work. Moreover, the paper fails to provide meaningful discussions or insights into the purported findings, which appear to be extrapolations from inconclusive experiments, thus leading to considerable confusion.
> Finding 1: The author tries different types of data and concludes that grounding data on natural images is the most important warm-up data for UI Grounding because it yields the best performance. Without additional insights, many different types of data and data combinations not yet explored by the author are possible. For example, perhaps [Gnd on Natural Images] + [QA on Natural Images] or [Gnd on Natural Images] + [QA on Natural Images] + [Text Localizations] combinations could perform even better. It would be beneficial if the author could provide more insights to guide future curation or selection of data. Without such insights, little new knowledge can be gained from these experimental results. Can the author explain why grounding with data from different domains can help? Why not warm up with data from the same domain?

> Finding 2: There is an additional dimension not ablated in Tab. 2. When using 2 or 3 stages of SFT with different data types, not only do the data types change, but the size of the data also changes. These two factors should be carefully segregated to draw conclusive results. For example, for r6, what happens if SeeClick data is also used for SFT-1 (a different set of data from SeeClick in SFT-2)?

> Finding 3: The data for complex UI-grounding is an order of magnitude smaller than others. Without using the same data size, it's unclear how reliable the conclusion is that we should "remain cautious of potential overfitting when fine-tuning with complex UI grounding data." Intuitively, I would expect that fine-tuning with data from the same domain should yield the best performance.

**Questions:**

1. Clarification on Fig. 4: Based on the descriptions in lines 355-366, the 5M results in Stage 1 should match the 212k results in Stage 2 and 625k results in Stage 3. However, in Fig. 4, why do the 5M results in Stage 1 (left) not match the 212k results in Stage 2 (middle) and the 625k results in Stage 3 (right)? Can the author kindly clarify?

---

### Official Review · Reviewer_JSx4 · 2024-10-23

**Soundness:** 3
**Presentation:** 4
**Contribution:** 3
**Rating:** 6
**Confidence:** 4

**Summary:**

The manuscript explores the design space for GUI-grounding vision-language models. It concludes that warming up with visual grounding tasks, using a simple-to-complex fine-tuning approach, scaling up data size, and utilizing effective image feature compression significantly enhance grounding accuracy for high-resolution UI images. These findings are supported by ablative experiments presented in the manuscript.

**Strengths:**

1. The manuscript is organized in a very clear and coherent manner.
2. The experimental results effectively validate the conclusions drawn.

**Weaknesses:**

1. Benchmark Selection: The manuscript focuses solely on grounding tasks for evaluation. While the primary contribution is exploring the design space for GUI-grounding MLLMs, it is essential to evaluate the grounding task within the context of an agent system. Improvements in a single component (e.g., the grounding part) do not necessarily translate to performance gains in the overall system, i.e., an autonomous UI agent that fulfills user intentions.

2. Comparison with Specialists: There is a need for comparisons with other specialized models.

3. Comparison with Other Base Models: Comparisons with other base models, such as InternLM, are necessary. It remains unclear whether techniques like visual grounding warm-up or feature compression are compatible only with LLaVA pre-trained representations or if they are generalizable across different base models.

**Questions:**

Please see the weakness section.

---

### Official Review · Reviewer_ZrzV · 2024-10-29

**Soundness:** 2
**Presentation:** 2
**Contribution:** 2
**Rating:** 3
**Confidence:** 5

**Summary:**

This work aims to develop Vision-Language Models (VLMs) with robust UI grounding capabilities. The paper presents the following key experimental findings:
1. Incorporating a warm-up fine-tuning step with visual grounding tasks enhances model performance.
2. a simple-to-complex fine-tuning curriculum maximizes data utility, and scaling up the warm-up dataset further improves grounding accuracy.
3. among various image feature compression strategies, a convolution-based compressor is found to be the most effective.
Based on these findings, the authors develop UI-Pro—a VLM fine-tuned to achieve strong grounding capabilities.

**Strengths:**

This work explores various design choices and configurations on both model and data for GUI grounding through comprehensive experiments and ablations, providing valuable insights for future research on enhancing GUI-based multimodal models. These findings contribute significantly toward advancing the development of effective UI agents.

**Weaknesses:**

1. The primary concern with this paper is that, while it presents several experimental findings specific to UI grounding, it lacks innovative design elements compared to existing vision-language models.

2. Although UI grounding is important, it is only one aspect of UI agent functionality. UI navigation, which requires both grounding and action selection, is more critical for UI applications. By focusing solely on grounding, this paper limits its broader applicability.

3. The paper makes several assumptions that raise concerns about the generalizability of its findings. The experiments are based on a specific VLM model (e.g., LLaVA), and it is unclear if these findings would hold for other VLM architectures.

4. The paper does not cite or discuss Ferrt-UI, despite Ferrt-UI's use of local patches, which is conceptually similar.

5. The model’s performance does not achieve state-of-the-art results.

6. Regarding Finding 1, most current state-of-the-art VLMs (e.g., LLaVA-OV, Qwen-VL2, Phi-3.5-V) already demonstrate strong text-rich visual perception capabilities. This suggests that the proposed warming-up strategy may be unnecessary.

**Questions:**

1. In Table 2, when focusing on UI grounding performance on ScreenSpot, a zero-shot benchmark, there appears to be no significant difference among r4, r6, and r8.

---

### Official Review · Reviewer_JXwL · 2024-11-09

**Soundness:** 2
**Presentation:** 2
**Contribution:** 2
**Rating:** 5
**Confidence:** 4

**Summary:**

This paper presents a UI-Pro, a vision-language model (VLM) designed to enhance autonomous interaction with user interfaces. It addresses the core challenge of UI element grounding, which involves accurately identifying and interacting with UI elements like buttons and text fields based on contextual references. UI-Pro achieves improvements  on grounding across benchmarks with fewer parameters than other leading models.

**Strengths:**

1. The validation of results across multiple benchmarks, demonstrating that the proposed methods lead to substantial improvements in grounding accuracy compared to existing models.

2. A detailed comparison of different data scaling and compression methods, providing clear evidence of their impact on model performance.

3. The use of a warming-up phase with general visual tasks before fine-tuning on UI-specific data is a strategic innovation that enhances adaptability and performance.

**Weaknesses:**

1. Limited Exploration of Model Variants in Pre-Training and Fine-Tuning:  The study could strengthen its contributions by including a comparative analysis of different pre-training tasks (e.g., using textual data combined with visual data, synthetic GUI elements).

2. While UI-Pro is tested across multiple benchmarks, these datasets are primarily focused on GUI scenarios without covering diverse UI types (e.g., industrial control interfaces, mobile applications, and accessibility-focused UIs).

3.The paper cautions against overfitting when scaling data in fine-tuning, yet it lacks in-depth analysis or metrics on model stability over different scales.

4.The paper achieves competitive results with fewer parameters than other models, but it does not address the computational requirements and inference efficiency (e.g., time and resource cost) of deploying UI-Pro on real-world applications, especially those with limited processing power or memory.

5.The paper demonstrates state-of-the-art grounding accuracy but does not include an in-depth error analysis, which would be valuable for understanding where the model's grounding might fall short.

**Questions:**

See details in 'Weakness' section.

---

### Note · Authors · 2024-11-15

I have read and agree with the venue's withdrawal policy on behalf of myself and my co-authors.